# Human T-Cell Leukemia Virus Type 1 Oncogenesis between Active Expression and Latency: A Possible Source for the Development of Therapeutic Targets

**DOI:** 10.3390/ijms241914807

**Published:** 2023-09-30

**Authors:** Francesca Marino-Merlo, Sandro Grelli, Antonio Mastino, Michele Lai, Paola Ferrari, Andrea Nicolini, Mauro Pistello, Beatrice Macchi

**Affiliations:** 1Department of Chemical, Biological, Pharmaceutical and Environmental Sciences, University of Messina, 98166 Messina, Italy; fmarino@unime.it; 2Department of Experimental Medicine, University of Rome “Tor Vergata”, 00133 Rome, Italy; grelli@med.uniroma2.it; 3The Institute of Translational Pharmacology, CNR, 00133 Rome, Italy; antonio.mastino@ift.cnr.it; 4Retrovirus Center and Virology Section, Department of Translational Research, University of Pisa, 56100 Pisa, Italy; michele.lai@unipi.it (M.L.); mauro.pistello@unipi.it (M.P.); 5Unit of Oncology, Department of Medical and Oncological Area, Azienda Ospedaliera—Universitaria Pisana, 56125 Pisa, Italy; p.ferrari@ao-pisa.toscana.it; 6Department of Oncology, Transplantations and New Technologies in Medicine, University of Pisa, 56126 Pisa, Italy; 7Department of Chemical Science and Technology, University of Rome “Tor Vergata”, 00133 Rome, Italy; macchi@med.uniroma2.it

**Keywords:** HTLV-1, adult T-cell leukemia, viral oncogenesis, Tax, HBZ, apoptosis

## Abstract

The human T-cell leukemia virus type 1 (HTLV-1) is the only known human oncogenic retrovirus. HTLV-1 can cause a type of cancer called adult T-cell leukemia/lymphoma (ATL). The virus is transmitted through the body fluids of infected individuals, primarily breast milk, blood, and semen. At least 5–10 million people in the world are infected with HTLV-1. In addition to ATL, HTLV-1 infection can also cause HTLV-I-associated myelopathy (HAM/TSP). ATL is characterized by a low viral expression and poor prognosis. The oncogenic mechanism triggered by HTLV-1 is extremely complex and the molecular pathways are not fully understood. However, viral regulatory proteins Tax and HTLV-1 bZIP factor (HBZ) have been shown to play key roles in the transformation of HTLV-1-infected T cells. Moreover, several studies have shown that the final fate of HTLV-1-infected transformed Tcell clones is the result of a complex interplay of HTLV-1 oncogenic protein expression with cellular transcription factors that subvert the cell cycle and disrupt regulated cell death, thereby exerting their transforming effects. This review provides updated information on the mechanisms underlying the transforming action of HTLV-1 and highlights potential therapeutic targets to combat ATL.

## 1. Introduction

The human T-cell leukemia virus type 1 (HTLV-1), also known as human T-lymphotropic virus type 1, was the first human retrovirus discovered. It belongs to subtype Delta of the subfamily *Orthoretrovirinae* and is endemic to the southwestern part of Japan, South America, the Caribbean, the Middle East, Australo-Melanesia [1], and Western and sub-Saharan Africa [2]. HTLV-1 has been recognized as the causative agent of adult T-cell leukemia/lymphoma (ATL), HTLV-1-associated myelopathy/tropical spastic paraparesis (HAM/TSP), and a number of inflammatory diseases. ATL is an aggressive malignancy that usually occurs in approximately 5% of infected adult individuals, 30–50 years after HTLV-1 infection. The prognosis of the aggressive subtype of ATL is very poor, with median survival ranging from 6 to 24 months. It is estimated that 5–10 million people are infected with HTLV-1 worldwide, although this number is most likely underestimated due to the lack of broader epidemiological studies. In vivo human-to-human transmission occurs through breastfeeding, blood transfusions, needle sharing, and sexual intercourse [3,4]. In addition, the rapid spread of HTLV-1 has been demonstrated in transplant recipients [5]. To understand the HTLV-1 infection/transformation that occurs in vivo, several noteworthy questions need to be addressed. The clarification of these questions may provide a representation of the complex interplay between HTLV-1 and the host gene expression that characterizes leukemogenesis and viral persistence in ATL [6]. This review provides detailed information on the current knowledge of the interplay between oncogenic viral proteins and cellular factors underlying the transforming action of HTLV-1 and provides a brief overview of the potential therapeutic targets arising from this interplay and possible future therapeutic approaches.

## 2. HTLV-1 Genome and Virus Transmission and Spread

The HTLV-1 genome consists of a small positive ss(+) RNA with a size of approximately 9 kB. After infection, HTLV-1 is integrated into the host DNA as a provirus, similarly to HIV-1. Both the sense strand and the anti-sense strand of the integrated provirus can be transcribed [7]. The viral genome is flanked by long terminal repeats (LTRs) at both the 5′ and 3′ ends. These direct repeats consist of three regions: the unique 3 (U3), the repeated (R), and the unique 5 (U5) region. The LTRs contain important elements necessary for viral transcription, polyadenylation, and integration [8]. The HTLV-1 genome includes the standard structural and enzyme genes *gag*, *pro*, *pol*, and *env*, which encode proteins essential for viral replication [9], as well as the coding potential for the accessory/regulatory proteins Tax, Rex, p12, p13, and p30 [10,11,12,13]. All these proteins are encoded by the sense strand, whereas the regulatory protein HTLV-1 bZIP factor (HBZ) is encoded by the anti-sense strand. HTLV-1 is suitable to be studied by means of in vitro models of infection. Because HTLV-1 is strongly cell-associated, unlike HIV-1, cell-free transmission is very limited in favor of transmission by cell-to-cell contact. The cell-to-cell transmission of HTLV-1 is facilitated by several transport strategies that allow the virus to spread, bypassing host immune responses. These strategies include transcytosis through epithelial barriers [14] and induction of membrane structures, such as virological synapses [15], cellular conduits [16], biofilm-like extracellular viral assemblies [17], and extracellular vesicles [18,19]. Viral infection in vitro usually requires cocultivation between the recipient lymphocytes and irradiated infected cells in the presence of IL -2 as a growth factor. After several passages in vitro, the number of which varies depending on the different cocultures the newly infected cells can be immortalized, as indicated by the fact that they no longer require growth factors. During the period between infection and immortalization in vitro, different cells in the infected cultures undergo different and opposite fates simultaneously: some of them undergo strong proliferation and others undergo apoptosis. This might depend on the balance between the differential expression of viral and cellular proteins at the cellular level and the influence on the survival/death pathways induced by viral gene expression in the infected cells [4], as detailed in the next paragraph. Although this process has been described in detail in vitro, it is reasonable to assume that a similar pattern could occur in vivo. The spread of the virus in vivo, similar for all retroviruses, occurs through two distinct pathways: the infectious pathway, from cell to cell, and the mitotic pathway, once the HTLV-1 genome is integrated into the host cell genome [20]. It has been clarified that infectious spread persists in parallel with mitotic spread even during the chronic phase of HTLV-1 infection [21]. HTLV-1 primarily infects CD4+ T cells, the population that is preferentially selected after in vitro infection. The clonal expansion of HTLV-1-infected cells is influenced by the host immune response [22], and fluctuating levels of the virus LTR in vivo seem to indicate that the virus actively replicates during chronic infections [23]. Studies aimed at comparing the clone abundance and distribution of the provirus in CD4+ and CD8+ T-cell subpopulations in HTLV-1-infected individuals have shown that CD8+ T cells carried only 5% of the provirus load, whereas the provirus was present in a greater number of CD4+ clones [24]. Nevertheless, CD8+ T cells expand strongly in vivo, but their role has not been fully elucidated [24]. The mechanisms of cell-to-cell transmission of HTLV-1 are summarized in Figure 1. Essentially, ATL is the end result of selecting an infected clone, in most cases CD4+, from the many infected clones. The selected clone gradually acquires the characteristics of the transformed cells over many years. The mechanisms underlying this long and complex process have been unraveled in recent years, largely thanks to advances in genomics and single-cell technology. See the subsequent sections and reference [25] for further information.

## 3. ATL- and HTLV-1-Driven Transformation: Generalities

ATL develops in 5% of individuals infected with HTLV-1 and is characterized by lymphadenopathy, skin lesions, hypercalcemia, and severe involvement of organs, such as the central nervous system, lungs, liver, spleen, and bone marrow. Survival prognosis is poor and may be as short as 6 months. ATL patients respond poorly to traditional anti-leukemia chemotherapy. Rather, more favorable responses have been achieved by combined treatment with zidovudine (AZT)plus interferon (IFN) [26] or allogeneic hematopoietic stem-cell transplantation [27]. In addition, biologic therapy has relied on mogamulizumab, which targets chemokine receptor 4 [28], and the immunomodulator and antitumorallenalinomide [29], a human CD30-directed chimeric antibody bound to the microtubule-disrupting agent [30]; the EZH1 and EZH2 dual inhibitor valemetostat, which disrupts the hypermethylation of histone H3 lysine 27 (H3K27) and allows the re-expression of repressed genes [31]; and histone deacetylase (HDAC) inhibitors [32,33]. The results of the above studies, although not very informative, encourage finding new potential biological targets in the mechanism of viral transformation (see Section 6 and Section 7). Several studies have shown that the final fate of transformed T-cell clones infected with HTLV-1 is the result of a finely tuned regulation of viral gene expression. In particular, the balance between the two major oncogenic viral proteins Tax and HBZ determines the fate of HTLV-1 infection [34]. Therefore, a regulated interplay between cell survival and cell death plays a key role in the selection of malignant clones.

One of the most important factors in the transformation and pathogenesis of HTLV-1 is the multifunctional viral protein Tax. This protein has the ability to interact directly with a variety of cellular proteins, including transcription factors, cell signaling proteins, cell cycle regulators, apoptotic proteins, and DNA damage response factors. Relevant to the aim of this review is the assumption that Tax positively modulates the expression of Bcl3 through activating the phosphatidylinositol 3-kinase (PI3K)/Akt signaling pathway [35]. Even the sustained activation of the cellular transcription factor Nuclear Factor-κB(NF-κB) by Tax [36] adds to its multifunctionality towards cellular targets (see Section 4). In addition, Tax is known to play a role as a transactivator and regulator of transcription of HTLV-1 structural and enzymatic proteins and of its own transcription through interaction with the promoter within the 5′ LTR. Tax-mediated transactivation requires binding to a repeated sequence of 21 nucleotides rich in G/C, representing the Tax-responsive element (TRE) located in the U3 region of the 5′ LTR [37]. Tax recruits the cAMP response element-binding protein (CREB)/activating transcription factor (ATF) to the cyclic AMP response elements (CREs) [38]. This leads to the formation of a nucleoprotein complex that recruits other CREB-binding proteins (CBP) and p300 [39]. This multiprotein complex is a potent activator of the 5′ LTR promoter and of viral mRNA transcription. Even if initial studies were unsuccessful in detecting a direct interaction between Tax and host DNA, it was cleared that, when in complex with CREB, Tax actually binds to DNA on the 21 bp repeats [40,41]. On the other hand, Tax can also upregulate viral transcription by directly binding to an epigenetic repressor, histone deacetylase 1 (HDAC1). Indeed, Tax inhibits or dissociates the binding of HDAC1 to the HTLV-1 promoter, thereby regulating viral protein transcription [42,43].

## 4. Tax and Cell Signaling: Role of the Transcription Nuclear factor NF-κB

In addition to regulating viral transcription, Tax regulates the expression of several signaling molecules involved in the processes of oncogenesis, proliferation, immune response, and apoptosis. Among these, the nuclear transcription factor NF-κB plays a central role in coordinating the various cellular signals. The prototypical NF-κB complex corresponds to a heterodimer of the p50 (NFKB1) and RelA (p65) members of the NF-κB/Rel family of transcription factors [44]. Tax induces the phosphorylation and degradation of both IκBα and IκBβ, suggesting that this HTLV-1 regulatory protein can induce the nuclear translocation of NF-κB by acting upstream or at the level of IκB phosphorylation [45,46]. The activation of NF-κB is involved in a number of events involving cytokine production in inflammatory contexts. Evidence that Tax regulates the expression of multiple cytokines is that its inhibition is dependent on the inhibition of NF-κB in HTLV-1-infected cells [47]. The sustained activation of NF-κB has a remarkable impact on the immortalization and transformation of HTLV-I-infected cells. For additional mechanisms of Tax-dependent or Tax-independent activation of the NF-kB pathways during HTLV-1 infection, please refer to a recent review [48]. The reduction in Tax in primary ATL cells rapidly abrogates NF-κB activation, leading to the induction of apoptosis [49]. One distinctive feature of immortalized HTLV-1 cells is the robust expression of anti-apoptotic genes. It has been reported that the Tax-mediated upregulation of Bfl-1, a member of the Bcl-2 family, is strongly expressed in HTLV-1-infected T-cell lines [50]. Direct evidence for the role of Bcl-2 family genes comes from studies showing that the silencing of Bfl-1 and Bcl-xL decreased the survival of HTLV-1 cells [50]. In vitro studies have shown that Tax transactivates the anti-apoptotic survive in promoter via the activation of NF-κB [51]. Nevertheless, Tax has not been exclusively associated with the inhibition of apoptosis, but also with its activation (see Section 5).

## 5. Latency and Leukemogenesis: Role of Tax, HBZ, and Apoptosis

HTLV-1 infection can remain latent for years before full disease onset, suggesting that there are sophisticated mechanisms regulating the on/off switching of viral protein expression, but also the activation of genes related to the T-cell receptor/NF-κB and signaling related to immune surveillance, such as HLA and FAS [52]. In addition to the progression of infection, the expression of Tax is also critical for stimulating the cytotoxic T-lymphocyte immune response, which is thought to play a role in the viability of ATL long-term survivors [53]. The loss or reduction in Tax expression in immortalized cells protects them from immune response attacks and renders them more prone to survival, expansion, and proliferation through a process of continuous transient expression [54].

Tax is not the only factor responsible for the various changes observed in viral expression and cell fate during infection. As already indicated in Section 3, HBZ also plays an important role by counteracting Tax to some extent and, on the other hand, integrating the process of virus-induced leukemogenesis [55]. Initially, HBZ was not thought to directly impact HTLV-1-mediated immortalization; instead, it was considered to regulate the establishment and maintenance of chronic infection [56]. The expression of hbz gene has been shown to upregulate JunD abundance; HBZ heterodimerizes with JunD, in turn recruiting the transcription factor Sp1 to the 3′ LTR of the provirus to enhance its activity [57]. Successively, HBZ was shown to be indirectly involved in leukemogenesis by increasing the expression of two oncogenic miRNA, miR-17 and miR-21. These miRNAs, in turn, down-regulate the expression of the single-stranded DNA-binding protein hSSB2, thus promoting genomic instability [58]. Other mechanisms relevant to how HBZ can promote genomic instability include its capacity to induce the accumulation of double-stranded DNA breaks (DSBs) by attenuating the nonhomologous end-joining (NHEJ) repair pathway [59] and its ability to suppress the transcription of MutS homologue 2 (MSH2), an essential mismatch repair factor, by inhibiting nuclear respiratory factor 1 (NRF-1) [60]. Another indication of the possible involvement of HBZ in neoplastic transformation is the detection of its translocation to the nucleus in cells from ATL patients, but not in those from asymptomatic carriers or from patients affected by non-neoplastic pathologies, where HBZ is exclusively localized in the cytoplasm [61]. In fact, the nuclear localization of HBZ in ATL cells could favor its functions on the cellular gene promoters of infected cells or its interaction with the host transcription factors involved in the leukemogenesis process. Moreover, HBZ was shown to delay or prevent Tax-induced senescence by modulating the hyper-activation of NF-kB by Tax, thus facilitating HTLV-1-induced leukemogenesis [34]. An additional mechanism by which HBZ can promote leukemogenesis is the inhibition of regulated cell death (RCD). It was shown that HBZ inhibits both the intrinsic and extrinsic pathways of apoptotic RCD by suppressing the transcription of Bim and FasL via targeting FoxO3a [62] and that level of hbz transcription were associated with the expression of the apoptosis inhibitor surviving [63].

Other accessory proteins play a selective role in HTLV-1 leukemogenesis. The p30 counteracts viral transformation by inhibiting the export of tax/rex mRNA from the nucleus [64], while p13 is involved in transformation by increasing the intracellular content of reactive oxygen species [65]. On the other hand, p12 enhances STAT5 activation in transduced peripheral blood mononuclear cells, allowing them to proliferate even in the presence of a low IL-2 concentration [66]. Thus, a number of different mechanisms underlie HTLV-1 latency/leukemogenesis involving both viral and cellular signals, although the main players are Tax and HBZ proteins. In brief, Tax is involved in the initiation of immortalization/transformation, whereas HBZ is required for the maintenance of the transformed stage. Nevertheless, the modulation of Tax and hbz expression at different stages of infection appears to favor the establishment of a dynamic, rather than static, state of latency and persistence. Indeed, HBZ has been shown to inhibit viral sense transcription and favor the entry of HTLV-1 provirus into the latency phase [67,68,69]. Ex vivo experiments have shown that, after HTLV-1 provirus integration into the cell, its replication essentially consists of a series of successive phases of burst and reactivation alternating with viral latency controlled by Tax, with the expression of pro- and anti-apoptotic genes playing a central role over cell cycle gene expression [70]. Experimental in vitro models have investigated the role of Tax in the switch between life and death, ultimately leading to the selection of immortalized clones in the final phase of HTLV-1 transformation (see below). An important point in this context is that seemingly conflicting results suggest that the role of Tax in controlling apoptosis is not clear: several studies have supported an effect preventing apoptosis via HTLV-1 gene expression in infected cells and have shown reduced susceptibility to apoptosis induced by anti-Fas [71] and TNFα [72]. This resistance to apoptosis has been associated with the activation of NF-κB via the Tax protein in HTLV-1-infected cells [51] or with the Tax-induced repression of pro-apoptotic genes [73] or the expression of anti-apoptotic genes [74]. However, despite the evidence for the prevention of cell death during HTLV-1 infection, other studies reported that the viral protein Tax not only exerts a somewhat anti-apoptotic effect but is also responsible for promoting apoptosis in HTLV-1-infected cells under certain experimental conditions, as mentioned in the previous paragraph. Clear evidence for this phenomenon was first provided by a study reporting that the expression of Tax in an inducible system turned out to induce, rather than inhibit, cell death [75]. The same and other authors even provided mechanistic details for apoptosis induction by Tax expression. Tax-triggered death was shown to be: i. dependent on ICE protease [76], ii. related to the upregulation of the Fas ligand (FasL) gene [77], and iii. promoted by the nuclear expression of the CREB-binding protein (CBP)/p300-binding domain of Tax [78]. In another study, Tax was shown to induce cell death through the NF-κB-mediated activation of the TNF-related apoptosis-inducing ligand (TRAIL) [79].

Taken together, these data fit with the apparent paradox of a dual role of Tax in apoptotic cell death in the early phases preceding the completion of the transformation process triggered by HTLV-1. To shed light into this subject, some of us pioneered the experimental model of HTLV-1 infection in vitro several years ago. Our model consisted of a long-term culture of PBMCs from healthy donors after exposure to irradiated HTLV-1-infected cells as viral donors [80]. We felt that such a model was the best strategy to recapitulate in vitro what occurs naturally in vivo in HTLV-1-infected patients during the long period of immortalization and transformation that could lead to ATL. Indeed, the results of this study showed that the in vitro HTLV-1 infection of mononuclear cells induces a high rate of cell proliferation during the first weeks accompanied by a heightened susceptibility to RCD through apoptosis. Moreover, there is a progressive decrease in cell death rates observed during the long-term culture phase, ultimately leading to cell immortalization [80]. Additionally, in situ hybridization showed that the cells undergoing apoptosis were indeed the infected ones, with HTLV-1 proviral DNA, rather than residual uninfected cells [80]. Although this was observed in an entire population, we hypothesized that some infected cells died by apoptosis because pro-apoptotic signals prevailed in response to infection triggered by viral gene expression, whereas the infected cells that survived were protected by the prevalent expression of anti-apoptotic genes, which were also triggered by viral proteins. Based on our results, we suggested, for the first time, that the induction of massive apoptosis in response to infection might act as selective pressure for the emergence of anti-apoptotic, well-endowed infected clones prone to immortalization [80]. A more recent study, performed at the single-cell level in the MT-1 ATL cell line, attempted to further elucidate the opposing effects of Tax on cell death by highlighting a situation in which Tax expression affects susceptibility to apoptosis. The results showed that a balance between antiapoptotic and proapoptotic genes depended on the on/off switching of Tax expression in the cells used in this study. A high expression of Tax was preferentially associated with an antiapoptotic gene expression scenario, whereas a low or absent expression of Tax was associated with a greater susceptibility to apoptosis. This switch was continuously active in culture and due to the coexistence of different expanding clones [81]. To explain the dual role of Tax in ATL, when cells progress in the cell cycle despite a low Tax expression, the results of another study hypothesized that ATL cells acquire genetic/epigenetic alterations during the transformation process. These allow bypassing the Tax/NF-kB-dependent induction of senescence [82]. However, it should be considered that the above studies were performed on ATL cells where the HTLV-1 transformation process had already been completed. Thus, the role of Tax in these cells could explain what happens in the cells of ATL patients in an advanced phase of the disease to maintain the leukemic state. It does not describe the role of Tax in the long premalignant phase of HTLV-1 infection, characterized by the oligo-/polyclonal expansion of non-malignant HTLV-1-infected cells. This precedes overt ATL in patients and actually involves HTLV-1-driven oncogenesis.

A recent study investigated the association between the early transcription of the positive viral strand, viral burst, and expression of pro- and anti-apoptotic genes in two naturally infected T-cell clones from patients transduced with a Tax-responsive timer protein [70]. The results showed that anti-apoptotic genes were expressed during the early positive-strand virus burst, followed by a phase in which proapoptotic genes outlasted the virus burst [70]. Another study of naturally HTLV-1-infected Tax-expressing T-cell clones from patients showed that a high Tax expression occurred during the burst phase, immediately followed by a phase of Tax expression heterogeneity associated with a poor proliferation, slow cell cycle, and high susceptibility to apoptosis [83]. Conversely, Tax-expressing clones showed long-term increases in proliferation and decreases in apoptosis [83].

Interestingly, apoptosis may play a role in the awakening of the virus from latency. It has been provocatively hypothesized that, in cells that are prone to apoptotic RCD in response to infection, apoptosis itself may awaken viral latency and promote viral replication through the direct upregulation of caspase 9. This protein, in addition to having a proapoptotic effect, may form a complex with Sp1-p53 and activate viral LTR [84]. On the other hand, viral reactivation from latency could also be triggered by metabolic changes, such as hypoxia, which has been shown to increase the transcription of the HTLV-1 proviral plus strand; conversely, the inhibition of glycolysis and mitochondrial electron transport chain hinders the transcription of the proviral plus strand [85]. Interestingly, the results of these recent detailed studies appear to be essentially consistent with the observations reported above. Overall, the results of these studies suggest that both the apoptotic RCD signaling pathway and the metabolic pathway in HTLV-1 infection may represent potential targets for the development of molecules to reverse latency.

Simplified schematics of the complex and elusive mechanisms leading to immortalization and transformation processes induced in HTLV-1-infected cells are summarized in Figure 2.

## 6. Proposals for HTLV-1/ATL-Targeted Therapy

Although the mechanisms of HTLV-1 immortalization/transformation have been extensively studied, it has not been possible to find a unique, suitable candidate treatment. This may be due to the large number of targets that could potentially be hit. The disease starts after the specific HTLV-1 infection of CD4+ cells. Therefore, hypothetically, a wide range of potential targets could be found within the viral genome/proteins, as well as among the host cell genes transcribed during infection that are highly subjected to virus-induced dysregulation. In particular, the specific cell signaling activated by HTLV-1 in infected cells includes multifunctional factors/complexes that could serve as targets for pharmacological intervention. Accordingly, several inhibitors of host cell signaling have recently been proposed in in vitro/ex vivo studies as potential anti-ATL therapeutics. These include apigenin, as an inhibitor of the aryl hydrocarbon receptor and transcription factor that enhances the cytotoxicity of antiretroviral drugs, and dimethyl fumarate as an inhibitor of the so-called CBM complex [86,87]. Another potential cellular target that has been studied for several years is PI3K. PI3K is involved in several processes of oncogenesis and is preferentially expressed in hematopoietic cells [88,89]. It has been demonstrated that the process of multiple nucleation and cell proliferation in ATL strictly depends on changes in the PI3K cascade [90]. Recently, a PI3K-δ/AKT inhibitor, idelalisib, was shown to specifically decrease the proliferation of ATL cells in vitro [91]. In addition, a dual PI3K and HDAC inhibitor, CUDC-907, previously used in patients with hematologic malignancies, has been considered as a potential candidate for ATL treatment. It was found to exert a cytotoxic effect in HTLV-1-infected cells by inhibiting the phosphorylation of downstream PI3K targets, such as AKT, REL A, and p70 S6K, and by decreasing HSP90 chaperone activity in ATL cells in vitro. Moreover, in HTLV-1-infected cells, CUDC-907 induced the upregulation of caspases, concomitant with the down-regulation of anti-apoptotic gene expression and the suppression of NF-κB activation by inhibiting IKKα/ß phosphorylation [92]. The suppression of NF-κB signaling proved to be one of the most important means to counteract the growth of HTLV-1-infected cells. This was demonstrated in HTLV-1-infected ATL cells subjected to mono-treatment with butein, a polyphenol that possesses pro-apoptotic and anti-proliferative properties by down-regulating AKT, AP1, and NF-κB activation [93]. Given the signaling mediated by NF-κB, a suitable approach may be to combine an inhibitor of NF-κB signaling with a chemotherapeutic or antiviral drug to promote susceptibility to cell death. The results of a recent study conducted by some of us showed how the pharmacological inhibition of IκBα could enhance the pro-apoptotic effect of AZT in chronically infected HTLV-1 cell lines [94]. Using a similar approach, a combination treatment was reported to be the most fruitful approach for ATL treatment, even in the case of co-treatment with biological drugs. The combination of ruxolitinib, an inhibitor of the JAK/STAT pathway constitutively activated in HTLV-1-transformed T cells [95], and the Bcl-2/Bcl-xL inhibitor navitoclax showed antitumor efficacy in an additive/synergistic manner on IL-2-dependent ATL cell lines and ex vivo on lymphocytes from ATL patients [96]. Recently, a triple combination of NF-κB and PI3K inhibitors with an inhibitor of the oncogenic driver Bromine and Extra-Terminal domain (BET) motif family, involved in the down-regulation of MYC transcription, was used synergistically to achieve an antiproliferative effect in ATL cells in vitro and ex vivo [97]. To evaluate the response of ATL patients to treatment in vivo, it may be critical to determine the efficacy of therapy on viral replication in addition to its effects on cell survival. A recent study in ATL patients subjected to long-term therapy with AZT and IFNα demonstrated that the combination treatment resulted in a complete inhibition of RT activity, a reduction in two other virological parameters, and a dramatic change in clonality pattern, as observed in the short-term cultures of PBMCs from patients who responded to the therapy [98].

Studies conducted in ATL patients have shown that an important goal is to induce cells to undergo apoptosis or generally move toward RCD, in response to therapy. Among the various ways to undergo RCD, interaction with the autophagy pathway should be considered. Autophagy may actually play a dual role in RCD, either initiating cell death or maintaining it [99]. The autophagy protein Beclin-1 appears to be involved in maintaining the activation of the factors NF-κB and STAT3 in HTLV-1-transformed cell lines [100]. Therefore, in the case of HTLV-1 infection, it seems that the suppression of autophagy may be an appropriate approach to treat ATL. Indeed, autophagy appears to be a “self-feeding” mechanism in tumor progression that supports cancer cell growth. Recently, chloroquine/hydroxychloroquine was shown to inhibit autophagic flux in ATL cells ex vivo. The mechanism involved the accumulation of p47 together with the autophagic protein LC3IIand led to increases in IkBα, resulting in the inhibition of NF-κB activation and susceptibility to apoptosis [101]. On the other hand, small-molecule inhibitors of sirtuin 2, the nicotinamide adenine dinucleotide-dependent deacylase involved in the control of cell cycle modulation, inhibited the growth of patient ATL cell lines not only by inducing caspase-dependent and -independent cell death, but also by increasing autophagosome accumulation and inhibiting autophagosome degradation [102]. Therefore, in this case, the upregulation of autophagic flux in ATL cells appears to be associated with mechanisms that induce cell death. These seemingly contradictory results encourage further studies on the role of autophagy in HTLV-1 infection and on a possible pharmacological modulation of autophagy as a novel strategy to target ATL.

Potential targets could be identified not only in genes directly involved in cell signaling but also among regulators of gene expression. Recently, 12 miRNA associated with the regulation of key cell signaling genes in ATL cases have been identified, thanks to a bioinformatics approach [103]. Based on the seemingly conflicting results obtained in defining a satisfactory treatment for ATL, it seems plausible that combination treatments may be the most appropriate approaches to inhibit a complex network of cell signaling as induced by HTLV-1 infection. Indeed, such a therapeutic strategy is likely to be the best weapon to avoid the possible feedback control aimed at restoring leukemic cell survival.

As mentioned in the previous section, even latency reversion may represent a potential strategy for a novel HTLV-1/ATL-targeted therapy. Latency could possibly be reversed by activating the expression of viral antigens that are likely to be recognized by the immune system. This would allow the recruitment of effector cells capable of knocking out the virus. Recently, a treatment to reverse latency has been proposed by using the histone deacetylase inhibitors (HDACi) panobinostat and romidepsin [104]. These inhibitors were shown to repress the transcription of Tax and of Tax-targeted genes, although only slightly.

A summary of the main suggestions for HTLV-1-targeted therapy is provided in Table 1.

## 7. Potential of Gene Editing Technology in the Eradication of PersistentHTLV-1 Infection and ATL Therapy

CRISPR/Cas9 genome editing is a novel technology that uses a guide RNA (gRNA) to precisely cleave double-stranded DNA at a specific site. After cleavage, double-stranded DNA breaks in human cells are normally repaired by the error-prone NHEJ pathway [105]. This can lead to insertions and deletions that alter gene-reading frames, disrupt DNA regulatory motifs, or damage RNA structures [106]. CRISPR technology has the potential to be a therapeutic strategy for HTLV-1 disease, as demonstrated by the use of zinc finger nucleases (ZFNs) to disrupt LTR promoter function [107] and inhibit the proliferation of HTLV-1-positive cell lines [108]. However, CRISPR/Cas9 has advantages over ZFNs and other gene editing approaches, such as simplicity, cost-effectiveness, and efficiency, and has been shown to reduce ATL cell proliferation in vitro by targeting HBZ [109]. Since the advent of gene editing, several studies have focused on targeting the HIV-1 provirus. This work may shed light on the efficacy, safety, and limitations of these approaches for targeting HTLV-1 [110]. The goal of gene editing approaches against HIV-1 is primarily to remove proviral DNA from the host genome. This can be achieved by targeting both LTRs and causing their disruption, followed by excision of the proviral DNA from the host genome. However, effective delivery and low off-target effects are critical for successful application in clinical trials [111,112]. In this sense, CRISPR technology is a promising gene editing tool, but it has some drawbacks that need to be addressed before it can be effectively used for antiretroviral therapy. In vitro, CRISPR/Cas9 can remove the integrated HIV-1 genome from cellular DNA, but in vivo, off-target activity, gene rearrangements, target selection limitations, and a limited number of effective transporters complicate the process. To overcome these problems, multiple RNA guide structures should be introduced into a single cell to ensure the cleavage of the provirus. However, the use of multiple guide structures may increase off-target activity and unpredictable DNA rearrangements. One of our recent studies examined the fate of HIV-1 provirus and cellular repair mechanisms triggered by CRISPR/Cas9. The study was conducted in two parts: the first part examined the fate of HIV-1 provirus in 293T cells and the second part confirmed the results in a human T-cell leukemia line latently transduced with HIV-1-GFP and in T-cell leukemia cells infected with a clinical lymphotropic isolate of HIV-1. The study found that, after CRISPR-mediated LTR ablation, the excised HIV-1 provirus remains in the cells for an extended period of time and can circulate as single molecules or concatemers that remain as episomes in the infected cells [113]. Non-integrated HIV-1 is abundant in resting, non-proliferating CD4+ T cells and leads to de novo virus production after the exposure to cytokine of the resting cells. The results of this study raise concerns about the persistence of CRISPR/Cas9-excised proviral DNA in the absence of antiretroviral therapy [113].

Although retroviral proviruses are largely restricted to HIV-1, HTLV-1 offers more sites for gRNA targeting than HIV-1 due to its highly conserved viral genome with remarkable sequence homogeneity. Moreover, as shown in Figure 3, CRISPR/Cas9 can disable both latent and actively replicating HTLV-1 and abrogate the function or expression of viral Tax and HBZ, which are the main drivers of HTLV-1-mediated transformation and proliferation. Targeting the viral LTRs involved in viral genome integration and gene expression may allow the effective treatment of HTLV-1-infected individuals, asymptomatic carriers, and ATL and HAM/TSP patients. The careful design of gRNAs that disrupt two viral elements simultaneously can disrupt overlapping reading frames between HBZ and the 3’LTR and Tax and the 3’ LTR [110].

## 8. Conclusions

Unfortunately, although some progress has been made in slowing its progression, ATL is still an incurable disease. However, in recent years, considerable progress has been made in defining the mechanisms at the molecular and genetic levels associated with the events underlying the complex viral/cellular system that leads to the selection of the transformed clone determining the onset of ATL in HTLV-1-infected patients. The definition of these mechanisms has already led to the identification of new therapeutic targets and corresponding agents that can act on them. Therefore, this seems to be an indication of how new therapeutic strategies may be found to counteract ATL in the near future. Nevertheless, a precise prediction of if and when it will actually be possible to use therapeutic treatments that can prevent or control ATL and potentially transform HTLV-1 infection into a chronic disease is currently not possible.

## Figures and Tables

**Figure 1 ijms-24-14807-f001:**
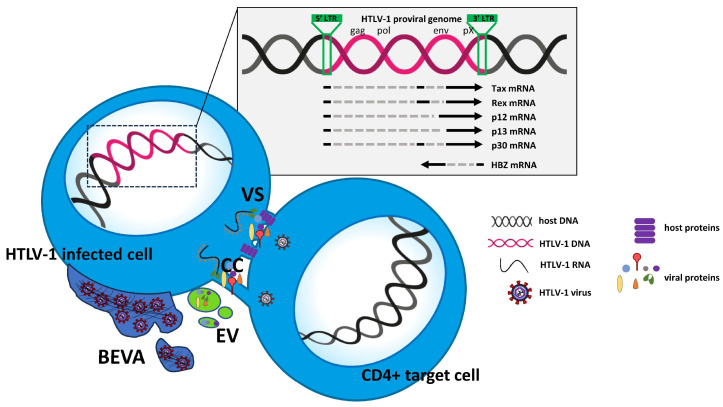
Schematic representation of HTLV-1 cell-to-cell transmission. VS = virological synapses. CC = cellular conduits. EV = extracellular vesicles. BEVA = biofilm-like extracellular viral assemblies. The mechanism of transcytosis is not reported due to difficulties in graphical representation.

**Figure 2 ijms-24-14807-f002:**
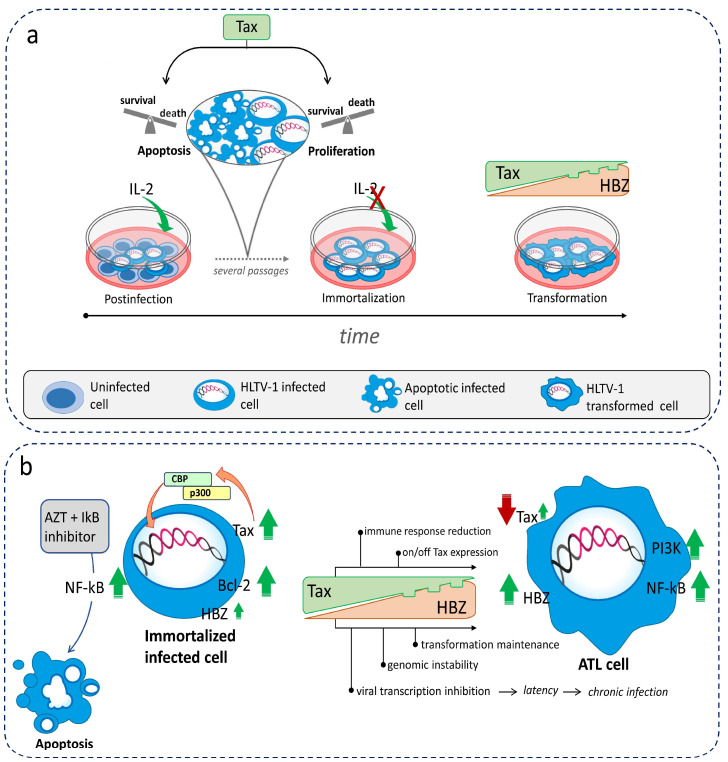
Schematic representation of (**a**) the outcome of HTLV-1 infection in vitro leading to immortalization/transformation and (**b**) the essential processes of leukemogenesis leading to ATL in HTLV-1-infected cells. The fragmented lower edge of Tax expression at low levels and the small green arrow for Tax in ATL cells in (**b**) show that Tax can be sporadically turned on and off during leukemogenesis.

**Figure 3 ijms-24-14807-f003:**
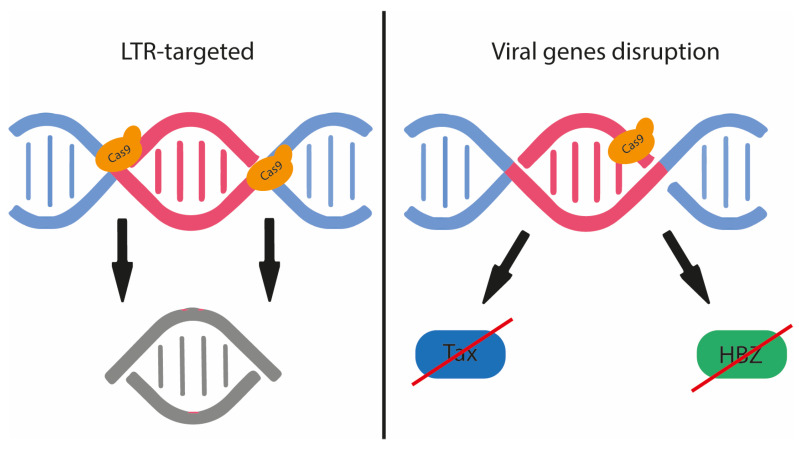
Schematic representation of how CRISPR-Cas9 gene editing could affect HTLV-1 integration and cell transformation.

**Table 1 ijms-24-14807-t001:** Proposals for HTLV-1/ATL-targeted therapy.

Proposed Therapeutic Treatment	Target		Available Results	
	Viral	Cellular	In Vitro/Ex Vivo	In Vivo
AZT + IFNα (98)	RT	IFN-receptor other? ^1^	Samples from patients: (a) Complete inhibition of RT activity and (b) reduction in virus parameters in responding patients; (c) Dramatic change in the clonality pattern.	Prolonged survival with respect to untreated patients
Idelalisib (91)	?	PI3K-δ/AKT	Inhibition of proliferation in ATL cells.	No
CUDC-907 (92)	?	PI3K/HDAC	(a) Induction of cytotoxicity in HTLV-1-infected cells; (b) Inhibition of HSP90 activity; (c) Increased caspase activity in ATL cells.	No
Butein (93)	?	AKT/AP1 NF-kB	(a) Induction of apoptosis; (b) Inhibition of proliferation of HTLV-1-infected and ATL cells.	No
AZT+ Bay 11-7085 (94)	RT-	IκBα phosphorylation	(a) Increased apoptosis; (b) Up-reg. pro-apoptotic and down-reg. anti- apoptotic genes in HTLV-1-infected/transformed cells.	No
Ruxolitinib+ Navitoclax (96)	?	JAK/STAT Bcl-2/Bcl-xL	Cytotoxicity in IL-2-dependent ATL cell lines and ex vivo in lymphocytes from ATL patients.	No
I-BET762+ Copanlisib+ bardoxolone methyl (97)	?	BET NF-κB PI3K	Inhibition of proliferation in ATL cells in vitro and ex vivo samples from patients.	Prolonged survival of ATL-bearing xenograft mice
Chloroquine/ Hydroxy chloroquine (101)	?	Autophagic flux	Ex vivo from ATL patients: (a) Inhibition of autophagy; (b) Accumulation of p47 with LC3IIand inhibition of NF-κB activation; (c) Proneness to apoptosis.	No
NCO-90/141 (102)	?	Sirtuin 2	(a) Increased apoptosis; (b) Autophagy in ATL cells.	No
? (103)	?	12 miRNA	In silico analysis identified 12 miRNA deregulated in HTLV-1 samples predicted to interact with 90 genes.	No

^1^ ? = unknown/not investigated.

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
