# Peer review of "Human T-Cell Leukemia Virus Type 1 Oncogenesis between Active Expression and Latency: A Possible Source for the Development of Therapeutic Targets"

_ijms, 2023, doi:10.3390/ijms241914807_

Round 1

Reviewer 1 Report (New Reviewer)

This review by Marino-Merlo et al. focuses on possible therapeutic targets to eradicate HTLV-1. The review encompasses several different areas of study and because of its lack of focus, references in some areas are superficial rather than comprehensive, and even completely lacking on some points. Eliminating paragraphs that are not the focus of the review would help strengthen the areas that the authors would like to review. For example, it would be beneficial to focus on the pathways shown in the Table 1.

Line 47: Prognosis of ATL is very poor (6-24 months): true for aggressive subtype of ATL, but not for smoldering and chronic.

Line 74: What does “HTLV-1 is a suitable model to study infection in vitro” mean?

There is a misunderstanding of the role of the extracellular vesicles (EV) in HTLV-1 infection. On line 80, and in Figure 1, the authors seem to suggest that the EV are responsible for transport of HTLV-1 virions. A careful reading of the article cited (Pinto et al, retrovirology, 2021) and the following article from the same group (Kim et al., Cells, 2021) do not conclude that the EV are carrying the virions but conclude that the EV are released before the virions to either help cell-to-cell contact or “prepare” recipient cells for infection. This should be corrected. In fact, a structure called “biofilm” has been shown to carry the virions (analyzed in several articles; reviewed in Maali et al., Front. Microbio; 2020).

Line 147: space needed between “of” and “21”.

Line 148: In fact, Tax binds directly to DNA when in a complex with CREB on the 21 bp repeats (Lenzmeier BA et al., Mol Cell Biol., 1998; Kimzey AL and Dynan WS, J Biol Chem, 1998).

Line 156: Please also cite Ego et al. Oncogene 2002, this work precedes the reference cited (38).

Paragraph 4: Please cite the review by Mohanty and Harhaj (Pathogens, 2020) for additional mechanisms on how Tax activates the NF-kb pathways (citations 40 and 41 are not sufficient and are old) and also Tax-independent mechanisms of NF-kb activation in ATL.

Paragraph 5: There is considerable work on the effect of HBZ (both the protein and RNA forms) on apoptosis that is not described in this paragraph (see articles from Matsuoka’s group). This omission is surprising since the title of the paragraph is “Latency and leukemogenesis: role of Tax, HBZ and apoptosis.” This paragraph also contains some random information on HBZ and doesn’t really address the role of HBZ.

Line 204: There are other mechanisms relevant to how HBZ promotes genetic instability: see Sakurada-Aono et al., Biochem Biophys Res Commun, 2023; Rushing et al. J Virol, 2018).

Line 204: Need to expand on this. Why is the presence of HBZ in nucleus an indication of its involvement in neoplastic transformation?

Line 218-219: There are additional references and mechanisms relevant to how HBZ represses HTLV-1 transcription in addition to ref 57. Please mention Gaudray et al., J Virol, 2002; Clerc et al., J Biol Chem., 2008.

Line 275: correct typo “clonesnt”

Line 313: The paragraph on HDAC could be move to next paragraph for clarity.

Figure 2: Why are p12 and p13 shown as increased in ATL cells? Why is NF-kB not shown as activated in ATL cells? Why is HBZ not shown in immortalized infected cells (recent article by Aristodemou et al., Plos Pathogens, 2023 shows HBZ expression as early as Tax expression).

Paragraph 6: Some of the targets described in this paragraph were not discussed earlier (PI3K pathway, JAK/STAT pathway). Why discuss the role of Tax on NF-kB but not on PI3K and JAK/STAT? This gives an overall feeling of confusion to the reader.

Table 1: Please put the references of the studies in the table.

Lane 362: typo “sing” instead of “Using”

Lane 421: Typo “Hb” instead of “Hbz”.

OK.

Author Response

We would like to thank all the reviewers for taking the time to review our manuscript a third time and for their comments and suggestions, which we believe have allowed us to further improve the manuscript. We have made the appropriate changes in the highlighted version of the revised manuscript as areas highlighted in yellow.

Answers to Reviewer 1:

General

This review by Marino-Merlo et al. focuses on possible therapeutic targets to eradicate HTLV-1. The review encompasses several different areas of study and because of its lack of focus, references in some areas are superficial rather than comprehensive, and even completely lacking on some points. Eliminating paragraphs that are not the focus of the review would help strengthen the areas that the authors would like to review. For example, it would be beneficial to focus on the pathways shown in the Table 1.

Answer: We thank the reviewer for his/her comments. We agree with the reviewer that our manuscript covers several different areas of the HTLV-1 study and that this may be a limitation as it may appear to lack focus. However, as clearly expressed in the title of the review and clarified in the response to the previous review, the particular focus of this review is to attempt to link basic research on the mechanisms of HTLV-1 leukemogenesis to the prospects for new therapeutics for ATL. Consequently, we were forced to include topics from different fields to support the main goal of our review! Nevertheless, we accepted the suggestion to do our best to avoid the lack of relevant references, as detailed in the “Specific Comments," by including 13 new references in the revised version.

Specific comments

Line 47: Prognosis of ATL is very poor (6-24 months): true for aggressive subtype of ATL, but not for smoldering and chronic.

Answer: Thanks for the comment. We specified this point in the revised version (see line 47).

Line 74: What does “HTLV-1 is a suitable model to study infection in vitro” mean?

Answer: Thanks. We have rephrased the sentence to clarify what we mean (see lines 74-75).

There is a misunderstanding of the role of the extracellular vesicles (EV) in HTLV-1 infection. On line 80, and in Figure 1, the authors seem to suggest that the EV are responsible for transport of HTLV-1 virions. A careful reading of the article cited (Pinto et al, retrovirology, 2021) and the following article from the same group (Kim et al., Cells, 2021) do not conclude that the EV are carrying the virions but conclude that the EV are released before the virions to either help cell-to-cell contact or “prepare” recipient cells for infection. This should be corrected. In fact, a structure called “biofilm” has been shown to carry the virions (analyzed in several articles; reviewed in Maali et al., Front. Microbio; 2020).

We thank the reviewer for the clarification on this point. According to the comment, the structure identified as biofilm-like extracellular viral assemblies (BEVA) has been included among the membrane structures involved in HTLV-1 cell-to-cell transmission in the text (see line 80) and in Figure 1. Moreover, a reference related to BEVA and the article from Kim et al. on EV, have been added in the revised version (ref. 17 and 19).

Line 147: space needed between “of” and “21”.

Answer: done.

Line 148: In fact, Tax binds directly to DNA when in a complex with CREB on the 21 bp repeats (Lenzmeier BA et al., Mol Cell Biol., 1998; Kimzey AL and Dynan WS, J Biol Chem, 1998).

Answer: Thanks for this clarification.  The revised text has been accordingly modified and the mentioned manuscripts have been included in the references (See lines 148 and 152-155, and ref 40 and 41).

Line 156: Please also cite Ego et al. Oncogene 2002, this work precedes the reference cited (38).

Answer: Done. See ref 43.

Paragraph 4: Please cite the review by Mohanty and Harhaj (Pathogens, 2020) for additional mechanisms on how Tax activates the NF-kb pathways (citations 40 and 41 are not sufficient and are old) and also Tax-independent mechanisms of NF-kb activation in ATL.

Answer: Done. The mentioned review was inserted into this part of the text, which was changed accordingly. (See lines 173-175 and ref. 48).

Paragraph 5: There is considerable work on the effect of HBZ (both the protein and RNA forms) on apoptosis that is not described in this paragraph (see articles from Matsuoka’s group). This omission is surprising since the title of the paragraph is “Latency and leukemogenesis: role of Tax, HBZ and apoptosis.” This paragraph also contains some random information on HBZ and doesn’t really address the role of HBZ.

Answer: Thank you for this comment, which helped us to improve the text. Indeed, we found that Section 5 did not adequately address the role of the HBZ. Therefore, this section has been substantially revised (see lines 199-201; 207-211; 213-224; related ref. 59, 60, 61, 34, 62, 63, 68, 69; see also next specific points).

Line 204: There are other mechanisms relevant to how HBZ promotes genetic instability: see Sakurada-Aono et al., Biochem Biophys Res Commun, 2023; Rushing et al. J Virol, 2018).

Line 204: Need to expand on this. Why is the presence of HBZ in nucleus an indication of its involvement in neoplastic transformation?

Line 218-219: There are additional references and mechanisms relevant to how HBZ represses HTLV-1 transcription in addition to ref 57. Please mention Gaudray et al., J Virol, 2002; Clerc et al., J Biol Chem., 2008.

Answer: All proposed articles were mentioned in the revised version. The text was changed accordingly (see lines 199-201; 207-211; 213-224; related ref. 59, 60, 61, 34, 62, 63, 68, 69).

Line 275: correct typo “clonesnt”

Answer: Done.

Line 313: The paragraph on HDAC could be move to next paragraph for clarity.

Answer. As suggested by the reviewer, the paragraph on HDAC in section 5 has been deleted and a new paragraph on this subject has been inserted at the end of section 6 (see lines 419-426).

Figure 2: Why are p12 and p13 shown as increased in ATL cells? Why is NF-kB not shown as activated in ATL cells? Why is HBZ not shown in immortalized infected cells (recent article by Aristodemou et al., Plos Pathogens, 2023 shows HBZ expression as early as Tax expression).

Answer: We thank the reviewer for the helpful comment. Figure 2 has been modified to reflect all points raised (note that in the schematic representation of the reciprocal expression of Tax and HBZ, the triangle of HBZ under Tax now begins at the left edge of Tax).

Paragraph 6: Some of the targets described in this paragraph were not discussed earlier (PI3K pathway, JAK/STAT pathway). Why discuss the role of Tax on NF-kB but not on PI3K and JAK/STAT? This gives an overall feeling of confusion to the reader.

Answer: Taking this comment into account, a paragraph on Tax's modulation of PI3K has been added to section 3 (see lines 139-142 and related ref. 35). Regarding the JAK /STAT pathway, we could not find any study that proves beyond doubt a role for Tax alone in its activation. Nevertheless, we added a new sentence in section 6 to specify that the JAK /STAT pathway is constitutively activated in HTLV-1-transformed cells (see lines 376-378 and associated ref 95).

Table 1: Please put the references of the studies in the table.

Answer: Done. The references of the studies cited in this review have been added to Table 1.

Lane 362: typo “sing” instead of “Using”

Answer: Done.

Lane 421: Typo “Hb” instead of “Hbz”.

Answer: Done.

Reviewer 2 Report (New Reviewer)

Thank you for writing such an important review article with brief but detailed description. 

Abstract is well written. 

Introduction and other sections are well described. 

Figures are self explanatory and we'll prepared. Please provide more illustrated figures for other sections if possible. 

Overall, excellent review article. Good luck. 

Author Response

We would like to thank all the reviewers for taking the time to review our manuscript a third time and for their comments and suggestions, which we believe have allowed us to further improve the manuscript. We have made the appropriate changes in the highlighted version of the revised manuscript as areas highlighted in yellow.

Answers to Reviewer 2:

General

Thank you for writing such an important review article with brief but detailed description. 

Abstract is well written. 

Introduction and other sections are well described. 

Figures are self explanatory and well prepared. Please provide more illustrated figures for other sections if possible.

Overall, excellent review article. Good luck.

Answer: We thank the reviewer for her/his positive comments and hope that this third revision has been further improved and meets his/her requirements.

Reviewer 3 Report (Previous Reviewer 1)

In the review entitled “Human T-Cell Leukemia Virus Type 1 oncogenesis between active expression and latency: a possible source for the development of therapeutic targets “Francesca Marino-Merlo and colleagues highlighted the oncogenic transformation process of HTLV-1 emphasizing the key role of latency and viral active expression. The authors also discussed the crucial role of HTLV-1 oncogenic proteins in the maintenance of persistent infection and indicated a few possible therapeutic approaches targeting the interplay of the virus and host factors.

However, the review lacks novelty and uniqueness as HTLV-1 induced cellular transformation process and current therapeutic strategies have been discussed in many other recent reviews with in-depth insights and proper discussion. There are several missing citations including Table 1 summarizing the HTLV-1-targeted therapy. The depiction of schematic illustrations is very poor and hard to understand. Additionally, this review needs significant grammatical editing as some of the sentences and headings are unclear and cannot be accepted in its current form.

Extensive editing of English language required

Author Response

We would like to thank all the reviewers for taking the time to review our manuscript a third time and for their comments and suggestions, which we believe have allowed us to further improve the manuscript. We have made the appropriate changes in the highlighted version of the revised manuscript as areas highlighted in yellow.

Answers to Reviewer 3:

General

In the review entitled “Human T-Cell Leukemia Virus Type 1 oncogenesis between active expression and latency: a possible source for the development of therapeutic targets “Francesca Marino-Merlo and colleagues highlighted the oncogenic transformation process of HTLV-1 emphasizing the key role of latency and viral active expression. The authors also discussed the crucial role of HTLV-1 oncogenic proteins in the maintenance of persistent infection and indicated a few possible therapeutic approaches targeting the interplay of the virus and host factors. 

However, the review lacks novelty and uniqueness as HTLV-1 induced cellular transformation process and current therapeutic strategies have been discussed in many other recent reviews with in-depth insights and proper discussion. There are several missing citations including Table 1 summarizing the HTLV-1-targeted therapy. The depiction of schematic illustrations is very poor and hard to understand. Additionally, this review needs significant grammatical editing as some of the sentences and headings are unclear and cannot be accepted in its current form.

Extensive editing of English language required.

Answer: We are sorry that this review does not meet with your approval. We hope that this further revision, which includes additional information and references compared with earlier versions, has increased the importance, novelty, and uniqueness of the article, and we hope that it is now suitable for publication in the International Journal of Molecular Sciences.

Specific comments

There are several missing citations including Table 1 summarizing the HTLV-1-targeted therapy. The depiction of schematic illustrations is very poor and hard to understand. Additionally, this review needs significant grammatical editing as some of the sentences and headings are unclear and cannot be accepted in its current form.

Extensive editing of English language required.

Following his/her comments and those of Reviewers 1 and 2, we have considered all of the above and the language has been extensively revised by a native English speaker with expertise in virology.

Round 2

Reviewer 1 Report (New Reviewer)

None.

None.

Reviewer 3 Report (Previous Reviewer 1)

The authors have included a few new important citations as per the recommendation. However, there is no significant improvement in the schematic illustration and figure legend explanation. Schematic illustrations are a crucial part of any good review article to summarize the complex physiological event in a simplified style which should be easy to understand and clear enough to attract the audience’s interest. The illustration of HTLV-1 cell-to-cell transmission depicted in Figure 1 could be found in other review articles with better quality and detailed explanations. Similarly, the Figure 2 schematic is visually misleading and poorly presents the on/0ff switch of Tax protein expression. As mentioned in the previous review comment, this review article also lacks uniqueness. Overall, this review article cannot be accepted in the present form for publication in the International Journal of Molecular Sciences.

Minor editing of English language required.

This manuscript is a resubmission of an earlier submission. The following is a list of the peer review reports and author responses from that submission.

Round 1

Reviewer 1 Report

Overall impression

In this study entitled, Human T-Cell Leukemia Virus Type 1 oncogenesis between active expression and latency: a possible source for the development of therapeutic targets“ Francesca Marino-Merlo and colleagues discussed the current understandings of interplay between oncogenic viral proteins and cellular factors with a brief overview of potential therapeutic candidates and possible therapeutic approaches targeting the interplay of the virus and host factors. However, the oncogenic mechanism of HTLV-1 as well as current therapeutic strategies has been discussed in many other recent reviews with in-depth insights with future prospectives. Hence, this particular review lacks uniqueness and an in-depth discussion of the functional regulation of oncogenic proteins of HTLV-1. Additionally, missing citations at various places with the poor depiction of schematic illustrations and presentation overall make it harder to understand from the reader's point of view.

Comments 

1-Missing citations at multiple places such as in lines 64-67, 70-76, `90-95, 103-108, 114-125, 129-130, 140-145, 152-155, 167-175, 194-202, 234-240, 252-259 and so on.

2- The authors discussed the pro-apoptotic role of Tax based on the study of transient expression of Tax which induces apoptosis or senescence in a few cell types. However, the complete dependency of ATL cells on Tax expression for its survival makes it controversial, and a proper infection model would be helpful to clarify the function of Tax in the context of apoptosis induction. Hence proper discussion of recent findings with appropriate citations is needed for a better presentation of the dual function of Tax. Additionally, authors should also discuss how infected cells counteract the Tax induced apoptosis/senescence to maintain persistent infection and survival.

3- The author should include a table to depict the possible drug targets for better clarity and understanding.

4- Schematic presentations are poorly depicted and lack clarity.

5- Missing citations are in multiple places in section 6 makes it hard to understand the background and experimental validation.

6- The conclusion or future prospects of the review is missing.

The language used is not sufficiently comprehensible and needs to be improved.

Reviewer 2 Report

Marino-Merlo et al. in their review titled, “Human T-Cell Leukemia Virus Type 1 oncogenesis between active expression and latency: a possible source for the development of therapeutic targets”, discuss the viral proteins and cellular events that directly contribute to oncogenesis in HTLV-1-infected T-cells. Throughout the body of the manuscript the authors detail the viral mechanisms associated with T-cell transformation, apoptosis regulation, and latency. The final two sections of the review provide a brief discussion on the currently characterized drug targets and gene editing technology to potentially combat HTLV-1 infection and disease development. Altogether, the manuscript serves to summarize the key viral events leading to cellular oncogenesis and the potential therapeutic options for HTLV-1-infected patients. Several comments are provided below to improve the quality of this manuscript:

Major Points:

1.     The grammar and English language used throughout the manuscript are not ideal and this makes the review difficult to read at some points. There are also multiple instances of unnecessary conjecture or hedge words included throughout the manuscript.  Notable examples are included below (note: more exist but this is a brief sample):

a.     Section 2 (lines 76-79): “In particular induction of a number of membrane structure allows a virus propagation avoiding host immune response, such as transcytosis into epithelial barriers [8], virological synapses [9], cellular conduits [10], extracellular vesicles [11].”

b.     Section 3 (line 106): “ATL is poor responsive to classical chemotherapy.”

c.     Section 4 (lines 140-142): “Besides regulation of viral transcription, Tax is able to regulate expression of a number of signalling involved in the processes of oncogenesis, proliferation, immune response, apoptosis,.”

d.     Section 6 (lines 294-297): “The network of peculiar cell signalling activated within HTLV-1 infected cells includes factors that are somehow interconnected each other and can act very likely as upstream or downstream targets of drugs.”

2.     Many sentences throughout the manuscript lack clarity and need to be rewritten in order to more clearly and accurately convey the information presented. These errors primarily include the use of ineffective language, improper terminology, run-on sentences, or poor sentence organization. Notable examples are included below:

a.     Abstract (lines 33-35): “This review will provide updated information on present knowledge on mechanisms underlying the HTLV-1 transforming action, also giving a concise overview of the potential therapeutic targets to contrast ATL.”

b.     Section 2 (lines 84-86): “In these lag time from infection to immortalization in vitro, cells undergo different, opposite fates, in the sense that they highly proliferate and likewise highly die by apoptosis.”

c.     Section 5 (lines 170-172): “The lost or decrease of Tax expression in immortalized cells rendered them protected by the attack of immune response and prone to expand and proliferate through a process of continuous bursting.”

d.     Section 6 (lines 346-347): “Autophagy is associated to a double face event regarding RCD, either as an inducer of death or as a counteracter of cell death [55].”

3.     Section 3 of the manuscript, titled “HTLV-1 immortalization/transformation”, does not provide enough discussion on the details of HTLV-1 transformation. The first paragraph focuses primarily on various therapeutic options for ATL, while the second paragraph summarizes how Tax regulates viral gene expression at the 5’ LTR.

4.     Figure 2 needs to be more clearly depicted. The schematic is somewhat disorganized and cluttered, and the graphics represented within the nucleus need proper labeling.

5.     There are several sections of the manuscript void of necessary citations including (but not limited to): lines 70-76, lines 89-95, lines 103-108, lines 116-126, lines 152-157, lines 167-176, lines 195-203, lines 288-299, lines 357-365, and lines 375-382.

6.     Page 2, line 64: The HTLV-1 genome is (+) ssRNA. It is not sense and antisense RNA. The (+) ssRNA genome is reverse transcribed into dsDNA. After integration, the viral genes can be expressed from the sense strand or the anti-sense strand of the provirus. 

Minor Points:

1.     Figure 1: Tax is not strictly decreased in ATL.  It can be silent (5’LTR hypermethylation, deletion, mutation), but it can also be sporadically expressed in an on/off manner.

2.     On lines 53-54, the authors characterized the cited study as “recent”, but the citation was published over 7 years ago.

3.     The following sentence is an oversimplification of HTLV-1 transformation that does not accurately summarize the process in vivo (lines 100-101): “Essentially, transformation in vivo is associated to CD4+ T clone proliferation since the great majority of ATL cases are CD4 positive.”

4.     On line 130, the authors define the protein “CREB” as “CRE-binding protein”. Since “CRE” was not yet defined at this point in the manuscript, the authors should alternatively describe this protein as, “cAMP response element-binding protein”. 

5.     Lines 154-157 include the following two sentences, “It has been reported that Tax mediated activation of NF-κB increased expression of antiapoptotic genes. This was revealed because of the high expression of a gene belonging to the Bcl-2 family, Bfl-1, that was highly expressed in HTLV-1 infected cells T cell lines in comparison with the uninfected ones.” The second sentence does not provide enough direct evidence to support the claim made in the first sentence. In addition, the second sentence needs to include a citation. 

6.     The following sentence is incorrect and does not accurately convey the findings of citation #25 (lines 178-179), “It has been demonstrated that HBZ controls its own expression by recruiting JunD within the 5’ LTR of the provirus [25].” The cited manuscript states that hbz gene expression upregulates JunD abundance and HBZ heterodimerizes with JunD to enhance its activity. Most importantly, Sp1 is primarily responsible for recruiting JunD to the 3’ LTR of the provirus, not the 5’ LTR.

7.     On lines 183-184, the authors state that, “The p30 counteracts viral transformation by inhibiting the export of Tax/Rex mRNA into the nucleus [27]”. This sentence is confusing and inaccurate since p30 inhibits the export of tax/rex mRNA out of the nucleus. Also, “tax/rex” should be italicized in this context.

8.     On lines 186-187, the authors state that, “p12 induces CD4+ cell proliferation through activation of the nuclear factor of activated T cells [29].” There is little-to-no evidence supporting this claim in citation #29, which details how p12 enhances the activation of signal transducers and activators of transcription 5 (STAT5).

9.     On line 205, “TNFa” needs to be rewritten as “TNFα”. 

10.  On line 376, the authors seemingly label “double-stranded DNA” as the acronym “DSB”. This acronym typically refers to “DNA double-strand break”. In fact, the authors use DSB in that respective context in the following sentence of the manuscript, “After DNA cleavage, DSBs in human cells are normally repaired through the error-prone non-homologous end joining pathway” (lines 376-378). To avoid potential confusion, the authors should more clearly designate the intended meaning of “DSB” on line 376.

See comments in review.